# Biomechanical Parameters that May Influence Lower Limb Injury during Landing in Taekwondo

**DOI:** 10.3390/medicina57040373

**Published:** 2021-04-12

**Authors:** Sihyun Ryu, Taek-kyun Lee

**Affiliations:** 1Department of Taekwondo, Korea National Sport University, Seoul 05541, Korea; hope222ysh@knsu.ac.kr; 2College of Education, Hankuk University of Foreign Studies, Seoul 02450, Korea

**Keywords:** Taekwondo, landing, impact, muscle strength, symmetry

## Abstract

*Background and Objectives*: The jumping kick of Taekwondo was a unilateral exercise that repeatedly moves in one direction. The exercise was exposed to the risk of injury due to the heavy ground reaction force and load in the landing. The first purpose of this study was to compare the impact force (IF), peak vertical ground reaction force (PVGRF), vertical loading rate (VLR), vertical stiffness (VS), and landing foot angle (LFA) during the landing of the jumping kick according to the experience of lower injury. The second purpose of this study was to investigate the lower extremities’ strength and the bilateral/ipsilateral asymmetry between the groups; *Material and Methods*: Ten injury-experience athletes (IG, age: 21.0 ± 0.8 years; height: 170.5 ± 4.1 cm; body weight: 66.7 ± 6.0 kg; career: 8.1 ± 5.0 years) and seven non-injury experience athletes (NG, age: 22.9 ± 4.0 years; height: 173.4 ± 3.1 cm; body weight: 67.9 ± 7.9 kg; career: 8.3 ± 5.0 years) participated; *Results*: There was no statistical difference between the two groups in the landing and lower extremity muscle strength impact variables. However, in the bilateral asymmetry of the ankle plantar flexor and the ipsilateral asymmetry of the hip abductor/adductor, IG was statistically larger than NG (*p* < 0.05). The landing foot angle also showed negative correlation to all impact variables (IF, PVGRF, VLR, and VS) (*p* < 0.05); *Conclusions*: It is desirable to place the landing foot down at a wide angle to prevent injury in performing Taekwondo jumping kicks. Although NGs have been shown to have better muscle strength symmetry and balance in some areas compared to IG, further research is needed to determine whether they are effective in preventing injury.

## 1. Introduction

A Taekwondo demonstration shows what Taekwondo is by performing the power and special skills polished through Taekwondo training, and this contributes to the spread and development of Taekwondo by inspiring people who have watched the Taekwondo [1]. Taekwondo demonstration skills were quantitatively developed over and over in the 2000s as a number of organizations were established. In contrast, the breaking skill, which accounts for about 70% of the total demonstration program, has been developed, while it has been modified into higher kicks, more kicks, and more turns [2,3]. In recent years, the target height and the number of kicks in the case of kicking skills in the air have increased and, simultaneously, problems and risks of landing have been reported [4,5,6]. In particular, jumping kicks are reported to have a high risk of injury because they have to make one leap, turn, and rotate in the air, and land after kicking multiple targets at higher positions [7,8]. Thus, the frequency of lower extremity injury is higher than other parts of the body [9]. In addition, three out of five Taekwondo athletes are reported to have experienced knee joint anterior cruciate ligament injuries at the moment of landing [10].

In a study [6] that investigated the effect of the single-leg landing during the jumping kick on the injury factors to the lower limb, it was reported that the lower joint angle and the lower muscle moment differed depending on a landing foot angle (LFA). It has been reported that additional abduction and internal rotation of the knee joint, as well as the lateral flexion of the trunk, can directly affect anterior cruciate ligament damage [11,12]. A study [13] that examined the impact reduction effect during jumping kick by inducing to change the LFA reported that turning the landing foot over 90 degrees and putting it down can reduce the impact force (14%), the peak vertical ground reaction force (8%), the vertical loading rate (23%), and the vertical stiffness (24%). LFA refers to the opening angle of the landing foot relative to the anterior direction of the center of mass (COM) during the jumping kick [13]. The impact force (IF) refers to the impact itself transmitted to the human body, and the vertical loading rate (VLR) refers to a variable predicting overused injury of the musculoskeletal system [14,15,16]. In addition, the vertical stiffness (VS) refers to an index representing efficient vertical movement during landing [17,18]. Therefore, it is necessary to compare the impact force and vertical loading rate at the landing after a jumping kick according to the lower limb injury experience and to examine the fundamental cause of lower limb injury.

In Taekwondo breaking technique, as the technical difficulty increases, there tends to create a one-directional movement because of one-sided performance skill. Unilateral exercises, such as tennis, badminton, and ping pong, mainly use one-sided muscle, which can cause left and right muscle imbalances and body changes [19,20]. In the case of athletes who repeatedly practice Taekwondo demonstration techniques, the soft tissue shortening or dynamic disorder involved in the range of motion may result, which may lead to sports injuries [21]. Previous studies have reported that lower extremity strength’s asymmetry is directly related to athlete injury [22,23]. A study on the lower extremity strength and asymmetry of female athletes participating in eight weight-bearing varsity sports and their association with injuries reports that the asymmetry of hip joint extension strength and knee joint flexural strength is more than 15% [24]. In addition, the asymmetry problem of lower extremity strength in various contact sport situations, including limited contact and non-contact, has been the major subject of recent research [25,26,27,28,29,30,31]. It has been reported that the rate of quadriceps and hamstring muscles in landing motion affects anterior cruciate ligament injury [32,33,34], and that hamstring muscles should be strengthened to prevent injury [32]. As such, in looking at the risk of injury during landing, it is necessary to investigate at the ratio of muscle strength between the agonist and antagonist muscle. Therefore, since the Taekwondo breaking technique is a unilateral exercise that repeatedly moves in one direction and exposes the individual to a heavy load in landing after performing the advanced technique in the air, examining the lower extremity strength of Taekwondo athletes and analyzing the symmetry index (SI) of bilateral (left vs. right)/ipsilateral (agonist vs. antagonist) muscular strength are tasks that must be performed for injury prevention and rehabilitation.

The first purpose of this study is to investigate the IF, peak vertical ground reaction force (PVGRF), VLR, VS, and LFA during the landing of Taekwondo jumping kicks according to lower injury experience. This study’s second purpose is to compare the lower extremity strength and the asymmetry of bilateral (left vs. right)/ipsilateral (agonist vs. antagonist) muscular strength between the groups. The first hypothesis is that the athletes who experience injuries (IG) show more significant impact variables (i.e., IF, PVGRF, VLR, and VS) and LFA than the athletes who do not experience injury (NG) during landing. This study’s second hypothesis is that the IG shows decreased low extremities’ strength than the NG. The third hypothesis is that the IG shows more significant bilateral/ipsilateral asymmetry than the NG.

## 2. Materials and Methods

### 2.1. Participants

An initial sample size of 14 was determined using the VLR of previous studies [13] concerning the impact variables during Taekwondo jumping kick landing. The mean and standard deviation of the VLR from the precedent mentioned above were used to calculate the effect size ($\eta_{p}{}^{2}$) of 0.226 with a statistical power set to 0.80 and an alpha level of 0.05 (G-power software). However, considering that group classification is made according to the experience of injury, 17 athletes with over 3 years of Taekwondo demonstration experience participated in the study. All participants were informed about the study’s purpose and contents through prior education, and the study was conducted after filling out a written consent form for athletes who voluntarily participated in the study. In addition, this study was approved by the Institutional Review Board, the IRB of K University (IRB Number: 1263-201810-HR-055-02). Research on injury experience and types were conducted in the same way as the National Olympic Committees (NOC) injury, whereas disease reports were used in the 2010 Vancouver Winter Olympics, the 2012 Summer Olympics in London, the Sochi 2014 Winter Olympics, and the 2016 Rio de Janeiro Summer Olympics [35,36,37,38]. Groups were divided into the non-injury experience group (NG, n = 7, age: 22.9 ± 4.0 years; height: 173.4 ± 3.1 cm; body weight: 67.9 ± 7.9 kg; career: 8.3 ± 5.0 years) and injury experience group (IG, n = 10, age: 21.0 ± 0.8 years; height: 170.5 ± 4.1 cm; body weight: 66.7 ± 6.0 kg; career: 8.1 ± 5.0 years). All of the IG suffered injuries during the landing in the same leg (left foot) after performing high-difficulty techniques, and most of the injuries were caused by sprains (n = 4) and ligament rupture (n = 6) in the knee and ankle joints.

### 2.2. Procedure

The jumping front kick was analyzed to examine the causes of injury to the jump kick’s landing movement in Taekwondo. First of all, sufficient warm-up was performed to induce the most natural motion and prevent injury, and 18 reflective markers (left and right acromion, left and right humerus lateral epicondyle, left and right ulnar styloid process, left and right iliac crest, left and right greater trochanter, left and right femoral condyles, left and right malleolus, left and right heel, and left and right 2nd phalanges) made with a diameter of 2 cm was attached to the center point of the joints. The kicking target height was unified to 230 cm. All participants performed jumping, kicking, and landing by the same leg (left foot). The same Taekwondo mats of the competitions were installed on the running, jumping, and landing floor. At this time, the pre-landing movements between the two groups were induced to be performed as identically as possible. The number of attempts was conducted five times, and the effect on the jumping kick was minimized with sufficient rest time. The starting position was pre-adjusted so that landing could occur on the force plate (AMTI BP 1200 × 1200, USA), whereas eight infrared cameras (Oqus 300, Qualisys, Göteborg, Sweden) were installed nearby. The infrared camera sampling rate was set to 200 Hz, and the force plate to 2000 Hz. Next, the lower extremity strength was evaluated in the maximum isometric strength and the strength measurement movements were the hip joint flexion, extension, abduction, adduction, external rotation, internal rotation, knee joint flexion, extension, dorsiflexion, and plantar-flexion as shown in Figure 1 [39,40,41]. At this time, the maximum isometric strength was measured using the manual muscle tester (Lafayette Hand-Held Dynamometer 01165, Lafayette, USA) for three seconds, and the average value of three measurements was used.

### 2.3. Data Processing

The ground reaction force data for the analysis of impact variables in this study were obtained with raw data from Qualisys Track Manager Software (Qualisys, Göteborg, Sweden), and the IF, PVGRF, VLR, VS, and LFA were calculated by Matlab R2009b Software (The Mathworks, Natick, MA, USA). At this time, ground reaction force data were noise-removed and smoothed using a Butterworth 2nd-order low-pass filter, and the cut-off frequency was set to 12 Hz.

### 2.4. Analysis Phase

In order to analyze the impact variables in the landing phase of the jumping kick according to the experience of lower extremity injury, the landing phase (i.e., the impact absorption phase) was set from the initial foot contact to the minimum height of the center of mass (COM) after breaking the 230 cm high target, as shown in Figure 2. The initial foot contact was defined as the time when the ground reaction force was appeared more than 10 N.

### 2.5. Analysis Variables

The IF, PVGRF, and VLR were calculated as shown in Figure 3a. First of all, the impulse was calculated by integrating the landing phase’s vertical ground reaction force, and the IF was the impulse divided by the time, which means the impact transmitted to the lower body when landing [42]. The VLR was PVGRF divided by the time, which is defined as the load rate transmitted to the human body [42]. At this time, the ground reaction force data were divided by body weight, and the static vertical ground reaction force was standardized to 1 to eliminate errors among the subjects. VS was calculated as seen in Figure 3b by the PVGRF at the landing and the vertical displacement of COM [17]. The LFA defined the foot segment opening angle for the jumping and landing direction when the jumping kick landing was completed (Figure 3c). At this time, the foot segment was defined as heel and 2nd phalanges of landing foot.
$$\mathrm{Impulse}=\int_{E1}^{E2}\mathrm{Vertical}\;\mathrm{GRF}\;\mathrm{dx},\;\mathrm{IF}=\frac{Impulse}{\Delta t\left(P1\right)},\;\mathrm{VLR}=\frac{PVGRF}{\Delta t\left(P1\right)},\;\mathrm{VS}=\frac{PVGRF}{vertical\;displacement\;of\;COM}$$

The SI for analyzing the bilateral (left and right)/ipsilateral (agonist and antagonist muscle) symmetry of the strength of the lower extremity was calculated as follows. As the SI was closer to 0%, it can be evaluated to be more symmetrical. It could range up to 200% [43].
$$\mathrm{SI}=\frac{\left|X_{L}-X_{N}\right|}{\frac{1}{2}\left(X_{L}+X_{N}\right)}\times 100\%$$

$X_{L}\;$ = lower body muscular strength of landing (or agonist), $X_{N}$ = opposite lower body muscular strength of landing (or antagonist muscle).

### 2.6. Statistical Analysis

In this study, an independent *t*-test was conducted to examine the differences between the groups. Normality checks were accomplished to confirm the normal distribution of dependent variables according to independent variables and, as a result, both Kolmogorov–Smirnov and Shapiro–Wilk satisfy the *p* > 0.05 and pass normality. For independent *t*-test results, 95% confidence interval and effect size (Cohen’s d) were presented. This study then performed a multiple regression analysis in whole sample (n = 14) with a stepwise method to examine the effects of the landing foot (i.e., LFA) and the lower extremities’ muscular strength on each impact variable (i.e., IF, PVGRF, VLR, and VS) and to investigate the cause of the injury in the landing motion of the jumping kick. The explanation power (r^2^) based on the independent variables was confirmed, and relative importance (β) was investigated. A statistical program (SPSS version 18.0; SPSS Inc., Chicago, IL, USA) was used to identify the differences at an alpha level of 0.05 for all variables.

## 3. Results

In this study, the impact variables (IF, PVGRF, VLR, and VS) and the LFA during the landing of jumping kick were investigated according to a Taekwondo demonstration player’s injury experience. The lower extremity strength and the bilateral/ipsilateral symmetry were compared. At first, the IF, PVGRF, VLR, VS, LFA, and the lower extremities’ muscular strength of the landing leg are shown in Table 1 and Table 2, and there was no statistical difference between the two groups. Next, the bilateral (left and right)/ipsilateral (agonist and antagonist muscle) symmetry indices were as shown in Table 3. The ankle plantar flexor SI of IG was greater than NG (11.5 ± 8.0% vs. 3.3 ± 1.6%, *p* = 0.018, d = 1.025, 95% CI −14.8, −1.6). The hip abductor/adductor SI of IG was greater than NG (13.3 ± 13.2 % vs. 6.3 ± 5.9 %, p = 0.011, d = 1.182, 95% CI −35.6, −5.3).

The examinations of the effect of the LFA and the lower extremities’ muscular strength on IF, PVGRF, VLR, and VS were as Figure 4. It was found that the coefficient of determination revealing the explanation power of independent variable (LFA) was indicated as R^2^ = 0.655, R^2^ = 0.225, R^2^ = 0.719, and R^2^ = 0.773, respectively, which was statistically significant at F (1, 15) = 28.441, 5.128, 38.421, and 51.212. The estimated multiple regression formula indicated as:IF = −0.015 × LFA + 4.501, PVGRF = −0.013 × LFA + 7.736, VLR = −0.889 × LFA + 250.170 and VS = −0.003 × LFA + 0.779.

LFA was revealed to have a statistically negative effect on IF, PVGRF, VLR, and VS.

## 4. Discussion

This study compared the impact variables (IF, PVGRF, VLR, and VS) and the LFA during the landing of jumping kicks based on the Taekwondo demonstration player’s injury experience. It investigated the lower extremity strength and the bilateral (left and right)/ipsilateral (agonist and antagonist muscle) symmetry.

Taekwondo jumping kick is the most basic of the demonstration’s skills, and all attention has been focused on target height. A related prior biomechanical study [44,45] has also focused on jumping and aerial motion. Recently, the risk of the landing of the athletes has been highlighted [5,6,7,8,9]. In particular, unstable landing motion has been reported to be capable of increasing injuries [6], and differences in impact variables (IF, VLR, and VS) depending on the foot segment angle at landing [13] were reported. In this respect, this study, which compares the impact variables of the landing phase between the experiences of the lower extremities injury and looks at the lower extremities muscle strength and symmetry index, which are directly related to the injury, is valuable in terms of providing immediate solutions to prevent injury.

There was no statistical difference between the impact variables, the landing foot angle, and muscle strength. A systematic review and meta-analysis evaluated 48 studies with a total of 5770 subjects and reported that 63% of individuals returned to preinjury activity levels after anterior cruciate ligament reconstruction surgery [46]. Moreover, a systematic review evaluated 10 studies with 312 patients and reported that the rate of return to preinjury activity level after syndesmotic ankle injuries was 93.8% [47]. As in the study above, no difference in the impact variables and the lower extremity strength was found between groups by performing the jumping kick after complete recovery from injury. Therefore, the first hypothesis that the IG would show a more significant impact variables than the NG and the second hypothesis that the IG would show decreased low extremities’ strength than the NG were rejected.

Looking at the PVGRF in this study, the force of approximately six times the body weight at the landing of the jumping kick was transferred to the human body (IG: 6.0 ± 0.7 BW, NG: 5.8 ± 0.7 BW). In a systematic review article [48], which analyzed PVGRF according to drop height during drop landing, more than six times the PVGRF was found for landing at a height above 50 m and barefoot landing was greater than shoe landing. Taekwondo jumping kick is considered very dangerous in terms of being barefoot and landing on a single leg. Therefore, it is necessary to consider a double leg landing that can disperse the human body’s impact and load [14,15]. Furthermore, as a result of the regression analysis looking at the effects on impact variables, only LFA affected all impact variables. It is considered effective to reduce impact by placing the foot down at a wide angle during the jumping kick landing, as the LFA had a highly negative correlation for the impact variables.

Next, the breaking technique mainly performed by the Taekwondo demonstration player is a representative unilateral exercise that requires one-way movement to be repeatedly performed. This study shows that the bilateral symmetry index of lower extremity strength is more than 10% and means that the left and right asymmetry levels are higher, and this asymmetry is considered a direct cause of musculoskeletal injury [24]. In this respect, it is considered to be of importance that the lower extremity strength of the Taekwondo demonstration player is measured and the bilateral (left and right)/ipsilateral (agonist and antagonist muscle) asymmetry is analyzed, as little analysis has been done so far. In particular, IG was significantly greater than NG in the bilateral symmetry index of the ankle joint plantar-flexion strength and the ipsilateral symmetry index of hip joint abductor/adductor. In relation to the Is mentioning that they all experienced injuries during the landing process, it is considered that the bilateral symmetry of the ankle joint movement and the ipsilateral symmetry index of the hip joint abductor/adductor are crucial factors in preventing injury at the moment of landing. It has been reported that the lower extremities’ muscle strength restoration and the symmetry of athletes who have an experience of lower-body injury were positively correlated with the symmetry of the landing motion [49].

Furthermore, softer landing motions were reported to absorb more energy from the hip joints than from the ankle joint [50], and the higher the landing height, the greater the muscle use around the hip joint [51]. Taekwondo athletes have a significant imbalance between knee joint extension and flexion muscles, so balancing the lower extremities’ muscles and strength is essential to promote their athletic performance and prevent injuries [52]. Considering the results of this study and the preceding study above, Taekwondo athletes should improve their muscle strength after lower limb joint injury, as well as their bilateral (left and right)/ipsilateral (agonist and antagonist muscle) asymmetry, similar to the preinjury period. Therefore, the second hypothesis that the IG would have more bilateral/ipsilateral asymmetry than the NG was partially accepted.

Based on this study’s results, it is effective to place the landing foot down at a wide angle to reduce the lower extremities’ injury at the landing of the Taekwondo jumping kick. Continuous management is also required to maintain the bilateral symmetry and ipsilateral balance of the lower extremities to prevent injuries of Taekwondo demonstration athletes. Research needs to look at the injury mechanisms and preventive strategies for various high-level Taekwondo skills in the future.

## 5. Conclusions

This study compared the impact variables of the landing phase in Taekwondo jumping kicks. It investigated the lower extremity muscle strength and the bilateral/ipsilateral symmetry of the landing phase in Taekwondo jumping kicks according to lower extremity injury experience. First, there was no statistical difference between the two groups in the landing and lower extremity muscle strength impact variables. However, in the bilateral asymmetry of the ankle plantar flexor and the ipsilateral asymmetry of the hip abductor/adductor, IG was statistically more extensive than NG (*p* < 0.05). The landing foot angle also showed negative correlation to all impact variables (IF, PVGRF, VLR, and VS) (*p* < 0.05). Therefore, it is desirable to place the landing foot down at a wide angle to prevent injury in performing Taekwondo jumping kicks. Continuous training for the symmetry and balance of lower extremity strength is expected to play an important role in preventing Taekwondo demonstration athletes’ injury. Further research is needed to determine whether muscle strength symmetry and balance are effective in preventing injury.

## Figures and Tables

**Figure 1 medicina-57-00373-f001:**
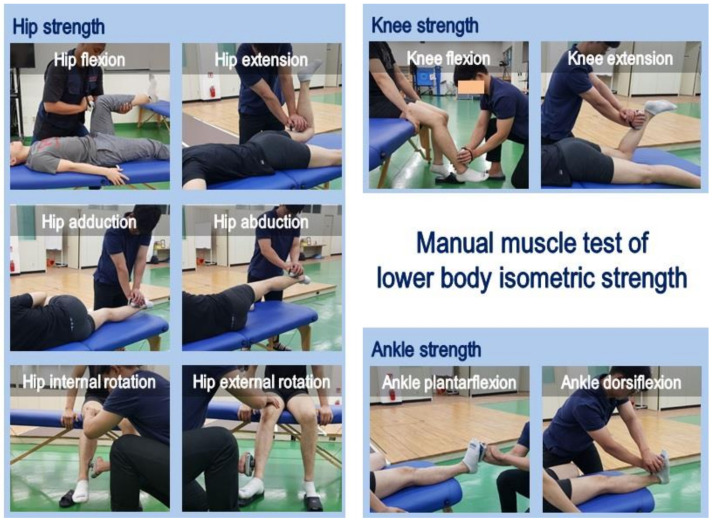
Manual muscle test of lower body isometric strength.

**Figure 2 medicina-57-00373-f002:**
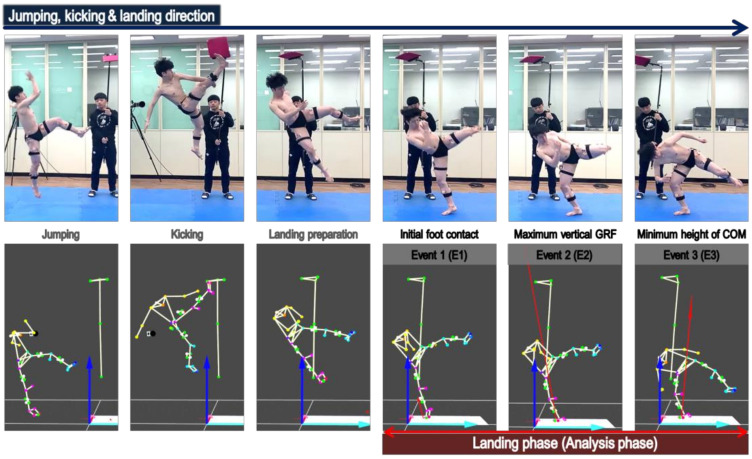
Experimental setup of jumping kick and analysis of the landing phase.

**Figure 3 medicina-57-00373-f003:**
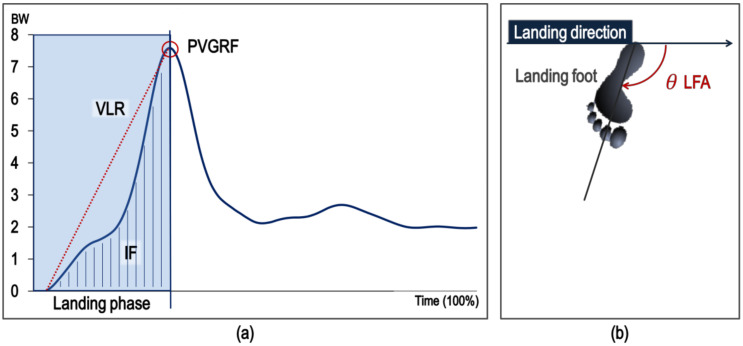
(**a**) Impact variables (impact force (IF), peak vertical ground reaction force (PVGRF), vertical loading rate (VLR)) of vertical GRF during landing, (**b**) landing foot angle (LFA).

**Figure 4 medicina-57-00373-f004:**
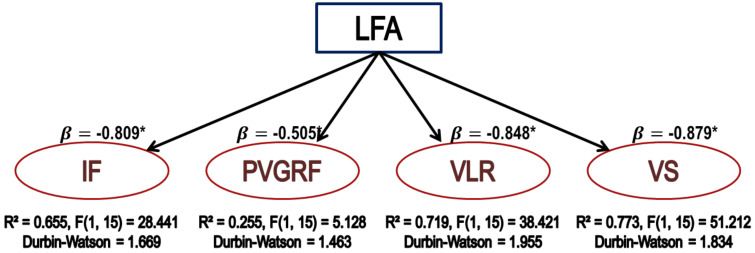
Description of the effect of the LFA on IF, PVGRF, VLR, and VS. * indicates statistical significance.

**Table 1 medicina-57-00373-t001:** Descriptive information for the impact variables and the landing foot angle.

Variables	NG	IG	95% Confidence Interval		*t*	*p*	Cohen’s d
			Lower	Upper			
IF (BW)	2.4 ± 0.5	2.6 ± 0.5	*−0.6*	0.4	*−0.547*	*0.593*	0.272
PVGRF (BW)	5.8 ± 0.7	6.0 ± 0.7	*−0.9*	0.6	*−0.564*	*0.581*	0.280
VLR (BW/s)	119.7 ± 24.4	137.0 ± 28.5	*−45.6*	11.0	*−1.302*	*0.213*	0.606
VS (BW/cm)	0.4 ± 0.1	0.4 ± 0.1	*−0.1*	0.0	*−1.306*	*0.211*	0.603
LFA (degree)	148.6 ± 24.8	126.0 ± 24.1	*−3.1*	48.2	*1.878*	*0.080*	0.935

NG: Non-injury group, IG: Injury group, IF: Impact force, PVGRF: peak vertical ground reaction force, VLR: Vertical loading rate, VS: Vertical stiffness, LFA: Landing foot angle.

**Table 2 medicina-57-00373-t002:** Descriptive information for the lower extremities muscular strength of landing leg.

(Unit: Nm/BW)							
Variables	NG	IG	95% Confidence Interval		*t*	*p*	Cohen’s d
			Lower	Upper			
Hip flexor	55.5 ± 13.4	65.3 ± 9.0	−21.3	1.8	*−1.794*	*0.093*	1.078
Hip extensor	64.0 ± 13.2	68.3 ± 15.9	−19.9	11.3	*−0.585*	*0.567*	0.270
Hip abductor	60.7 ± 12.4	72.1 ± 11.5	−23.8	1.1	*−1.941*	*0.071*	0.985
Hip adductor	60.4 ± 9.3	54.0 ± 10.0	−3.8	16.7	*1.346*	*0.198*	0.646
Hip external rotator	30.9 ± 7.1	31.0 ± 5.3	−6.5	6.2	*−0.051*	*0.960*	0.029
Hip internal rotator	29.5 ± 1.7	32.2 ± 6.5	−8.1	2.7	*−1.053*	*0.309*	0.411
Knee flexor	46.8 ± 6.3	47.3 ± 5.1	−6.4	5.4	*−0.179*	*0.861*	0.097
Knee extensor	67.4 ± 23.9	85.5 ± 17.1	−39.1	3.0	*−1.823*	*0.088*	1.057
Ankle dorsi-flexor	42.9 ± 6.3	49.6 ± 9.6	−15.5	2.2	*−1.598*	*0.131*	0.694
Ankle plantar flexor	99.1 ± 21.2	97.3 ± 18.8	−19.0	22.6	*0.184*	*0.857*	0.095

NG: Non-injury group, IG: Injury group.

**Table 3 medicina-57-00373-t003:** Descriptive information for the bilateral/ipsilateral symmetry index of the lower extremities muscular strength.

Variables	NG	IG	95% Confidence Interval		*t*	*p*	*Cohen* *’s d*
			Lower	Upper			
Bilateral symmetry index (%)							
Hip flexor	10.8 ± 6.7	10.1 ± 7.4	−6.7	8.2	*0.216*	*0.832*	*0.102*
Hip extensor	14.2 ± 17.8	9.2 ± 5.1	−7.5	17.6	*0.859*	*0.404*	*0.998*
Hip abductor	14.5 ± 14.3	8.5 ± 6.0	−4.7	16.7	*1.200*	*0.249*	*1.003*
Hip adductor	21.2 ± 20.7	13.6 ± 10.7	−8.7	23.9	*0.998*	*0.334*	*0.712*
Hip external rotator	14.8 ± 12.8	18.4 ± 18.6	−21.0	13.8	*−0.443*	*0.664*	*0.194*
Hip internal rotator	18.4 ± 18.3	14.1 ± 12.3	−11.5	20.0	*0.577*	*0.572*	*0.348*
Knee flexor	23.3 ± 20.6	10.1 ± 7.8	−1.9	28.3	*1.867*	*0.082*	*1.694*
Knee extensor	15.5 ± 13.2	12.7 ± 12.4	−10.5	16.2	*0.454*	*0.656*	*0.230*
Ankle dorsi-flexor	9.5 ± 4.8	10.7 ± 10.8	−10.6	8.1	*−0.292*	*0.774*	*0.119*
Ankle plantar flexor	3.3 ± 1.6	11.5 ± 8.0	−14.8	−1.6	*−2.649*	*0.018 **	*1.025*
Ipsilateral symmetry index (%)							
Hip flexor/extensor	11.3 ± 7.3	10.0 ± 7.0	−15.9	11.2	*−0.367*	*0.719*	*0.222*
Hip abductor/adductor	6.3 ± 5.9	13.3 ± 13.2	−35.6	−5.3	*−2.881*	*0.011 **	*1.182*
Hip external rotator/internal rotator	7.8 ± 6.4	11.2 ± 12.1	−9.2	15.0	*0.509*	*0.618*	*0.300*
Knee flexor/extensor	9.8 ± 6.8	25.2 ± 29.2	−27.3	13.9	*−0.690*	*0.501*	*0.387*
Ankle dorsi-flexor/plantar flexor	12.9 ± 15.2	18.6 ± 16.8	−9.7	37.0	*1.247*	*0.231*	*0.548*

* indicates statistically significant difference between the groups.NG: Non-injury group, IG: Injury group, M/L: Mediolateral, A/P: Anteroposterior.

## Data Availability

Not applicable.

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
