# Peer review of "Biomechanical Parameters that May Influence Lower Limb Injury during Landing in Taekwondo"

_medicina, 2021, doi:10.3390/medicina57040373_

Round 1

Reviewer 1 Report

Review of the paper medicina-1123774-peer-review-v1

The Authors undertake the biomechanical analysis of jumping kick in Taekwondo. Two groups: injury experience athletes and non-injury experience athletes participated in this research. The aim of this study was to investigate few biomechanical parameters which can influence the lower limb injury. The results showed no statistical difference between this two groups for force generated by examined muscle groups. The Authors proofed that asymmetry of force generated by the ankle plantarflexors and hip abductor/adductor was statistically larger in injury experience athletes group.

I really like the idea of the research, but the paper not polished is in many ways. There is no fluency in the transfer of knowledge and, most importantly, the content of the work does not correspond to the title. The authors have no recommendations on how to prevent injury. The only recommendation is to strengthen the muscles symmetrically, which is known to be difficult in asymmetric sports. I am curious why, with such advanced equipment, you do not analyze the muscle torques and angles during kick jump in phase of landing? In my opinion, only the platform would be sufficient for the analysis presented in the paper. Perhaps it would be worth analyzing other variables? The study groups are very small and it seems that you are not getting sufficient data. Additionally, the "injured" group is not well characterized. It also seems that the work should undergo linguistic correction. Please, see my comments below.

  • In my opinion, the title of the paper should be change. The Authors did not designed the strategy for preventing injury in Taekwondo. The article is rather a description and specification of parameters that should differentiate two groups of Teakwondo practitioners (injured and no injured). However, apart from the recommendation to maintain muscle balance and symmetry, I do not see others. Therefore, I suggest to change the title to something similar: Biomechanical parameters which may influence lower limb injury during landing in Taekwando.

Introduction:

  • Line 26-33 – six times the word “demonstration” is used. Please re-edit the content.
  • Line 34-35 - please specify this sentence (What kicks are you talking about?). Kicks can also be static without jumping.
  • In the introduction, there are parameters that can be understood in various ways. Therefore, please take a place in the introduction section to their explanation and give precise definition. I mean in particular: landing foot angle, impact force, vertical loading rate, vertical stiffness.
  • Line 49 – “… can reduce the impact force….” - please write by how many percent.
  • Line 56 – “… mainly use only one muscle” - this is a mental shortcut. Please edit sentence.
  • Line 61 – “ A study on the lower …” - Is this a general sentence for all sports or are you only referring to Taekwondo? Please edit sentence.
  • What do you mean: asymmetry of bilateral/ipsilateral muscle in the aim? This is not understandable, especially since this concept appears for the first time in the paper. Please re-edit the purpose of the paper.

Material and methods

  • It's good that the sample size was counted, but then unexpectedly the Authors from one group make two small groups. It seems that these numbers are too small to carry out complete statistics.
  • Line 97-102 - These sentences should be placed in line 89 - before the sentence “All participants were informed…”. There is no information about the injured group. When was the last injury before the tests? For what period of time the competitor was eliminated from sports activity? How was the treatment process going? Was the competitor fit on the day of the test? Were the players a homogeneous group in terms of injuries? It seems not. Sprains of the knee or ankle are completely different injuries than fracture. Please, provide a detailed description of the injury group.
  • On which limb did the athlets land on?
  • Procedure: Line 106 – 107 – “…reflective marker made with…” - was only one marker used in the research? Looking at figure 2 - probably no. Please write how many markers were on the human body, was it a full body diagram or just lower body? (It's worth putting the photo here). If the number of trials was 5 for one person - please indicate how many were taken for further analysis?
  • Line 115 – 120 - It is known that in order to make correct measurements of the force generated by the muscle groups, during the measurement, the remaining segments should be fully stabilized so that they do not help in movement and thus do not disturb the measurement. I do not see such items in Figure 1. Please explain. How many times have the measurements been taken and how long have the athletes held the maximum tension?
  • Line 133 - Ho did you define the initial contact? From the moment when first time GRF was appear?
  • Section 2.5 - In my opinion, this section should be written from the beginning. If the Authors will provide the mathematics formulas it will be easier to read. I also think that the units of each parameter should be given. Figure 4 and 5 is missing.
  • Line 146-147 – “The LFA defined the foot segment angle for the running and landing direction when the jumping kick landing was completed” - please describe exactly how it was counted. In base of which markers?
  • Figure 3: 1) You should not start your description from the point (a). The figure should have some general title. In (b) there is wrong citation – it should be in other style, like [], (c) – what does mean FA(+)?
  • Line 153- this information should be before (in the introduction section). Only here it was explained what is going on with SI.

Results

This section should be corrected. Such a large number of tables is not necessary. In my opinion, Authors should plot some results using MatLab and the subplot function. It will be more interesting for the reader. Table 4, 5, 6, 7 are not needed. All information are in the text.

Author Response

Thank you for your constructive comments on our manuscript. We have carefully revised the manuscript to improve the quality of the study. Please see our point-by-point responses below.

Reviewer 2 Report

General comments:

The article is professionally written, although some part is written in difficult consequences. Thus, the language might be improved in term of clarity. However, the topic and results are of a good interest. The statistical part should state little more about the n number used for regression. The weak part of the study is the n of participant, which is on the other hand understandable if elite sample is involved. The result section includes some inconsistency in referring regression to muscle strength and values in the reported tables. On the other hand, the discussion is the strong part of the study, however some more aspects of the results should be discussed. 

Specific comments:

Line 8, 14, 16, 21: The numbers in brackets after redundant, remove it.

Line 9: You should write heavy reaction forces, not just load.

Line 9 -13: this sentence is just too long. Split this sentence and revise it.

Line 20: What is relative relevance? This is in the context of abstract unclear, please revise this statement.

Line 21: This conclusion do not seems to be related to measurement outcomes. Revise it.

Line 24: some keywords are already in title, replace them.

Line 27 – 29: This sentence should be little edited for language clarity.

Line 86: the effect size should be written properly not in current form.

Line 115: The muscle strength measurement should have own paragraph or even sub-section. Moreover you should state the contraction duration and number off attempts for each muscle. Then in results report the ICC of this tests.

Line 138: Please explicitly define the impulse calculation. (Was it done to the peak values of different (e.g. 100%)?

Line 166: You should explicitly write that you run the regression in whole sample of n = 14 (not by groups).

TABLE 4 - 7 should be combined into one, where landing foot angle will be independent factor. Moreover, you should highlight what means negative or positive relationship, and the role  and individual b of muscle strength values.

Discussion,

This is a strong part of the article however, you should discuss following issues as well:

  1. Are there any other kinematic variable influencing dynamics such as https://pubmed.ncbi.nlm.nih.gov/33387442/
  2. Can you clearly state in results and discussion what is the role of isometric strength, kinematic and dynamic parameters. https://pubmed.ncbi.nlm.nih.gov/31531140/. You are stating at lines 195-197 this relationship but the referred Table 5 is not with muscle strength reported.

Did you actually included isometric Strength in the regression models?

Author Response

(The authors gave the same response as above.)

Round 2

Reviewer 1 Report

The Authors have significantly improved the paper and in my opinion it is now much more readable and transparent. I have no further comments and I recommend the manuscript for publication in its current form.